# GCMS profiling of bioactive phytocompounds from *Curculigo orchiodes* Gaertn. root extract and evaluation of antioxidant, and antidiabetic activities: A computational drug development approach

**Imran Mahmud**[1,2], **Md. Khalid Saifullah**[2], **Md. Niaj Morshed**[3], **Md. Arju Hossain**[4], **Naznin Shahria**[2], **Apurba Kumar Barman**[5]*, **Famim Ahmed**[2], **Md. Jakaria Islam**[6], **Toufiq Ejaj Khan**[7], **Nripendra Nath Biswas**[7]*

1 Department of Life Technologies, Biotechnology Division, University of Turku, Turku, Finland, 2 Department of Pharmacy, Khwaja Yunus Ali University, Sirajganj, Bangladesh, 3 Department of Biopharmaceutical Biotechnology, College of Life Sciences, Kyung Hee University, Global Campus, Yongin-si, Gyeonggi-do, South Korea, 4 Department of Biochemistry and Biotechnology, Khwaja Yunus Ali University, Sirajganj, Bangladesh, 5 Department of Pharmacy, R. P. Shaha University, Naryanganj, Bangladesh, 6 Department of Pharmacy, State University of Bangladesh, Dhaka, Bangladesh, 7 Pharmacy Discipline, Life Science School, Khulna University, Khulna, Bangladesh

* apurba_phr@rpsu.edu.bd (AKB); nnathbiswas@gmail.com (NB)

## Abstract

*Curculigo orchioides (C. orchioides),* a traditionally valued medicinal plant, has been utilized for centuries in the management of various ailments, but its full spectrum of therapeutical potentials remains underexplored. This study aimed to perform GC-MS profiling of bioactive phytochemicals as well as to evaluate the antioxidant and anti-diabetic properties of the ethanolic root extract of *C. orchioides* (ERCO) through an integrative approach combining *in vitro*, *in vivo*, and *in silico* methods. Phytochemical screening confirmed the presence of some major bioactive phytochemical groups including alkaloids, flavonoids, tannins, saponins, and steroids which are well-known for their pharmacological relevance. Antioxidant activity was demonstrated through high levels of total phenolic content (TPC), total flavonoid content (TFC), total tannin content (TTC) determined as 44.055 mg GAE/gm, 0.6768 mg QE/gm, and 103.375 mg TAE/gm of dry weight extract, respectively, along with notable ferric reducing antioxidant power (FRAP). Anti-diabetic potential was supported by significant *in vitro* inhibition of pancreatic $\alpha$-amylase and $\alpha$-glucosidase enzymes, with $IC_{50}$ values of 84.17 µg/mL and 36.33 µg/mL, respectively. *In vivo* studies in alloxan-induced diabetic mice further validated the extract's substantial blood glucose-reduction abilities (47.28% and 52.11% at the dose of 100 mg/kg and 200 mg/kg body-weight, respectively), indicating the potential for blood sugar regulation. GC-MS profiling confirmed the presence of 23 major phytochemicals, which were subjected to molecular docking studies against human glutathione peroxidase, peroxiredoxin 5,

**Data availability statement:** All relevant data are within the manuscript and its Supporting information files.

**Funding:** The author(s) received no specific funding for this work.

**Competing interests:** No authors have competing interests.

Catalase, sulfonylurea receptor 1 (SUR1), α-amylase, and α-glucosidase. Among them, 2-epoxy-3,4-dihydroxycyclohexano[a]pyrene (CID: 41322) emerged as a lead compound, exhibiting strong binding affinities for both α-amylase (−9.1 kcal/mol) and α-glucosidase (−8.8 kcal/mol). ADMET predictions and stable molecular dynamics simulation outcomes further underscored its drug-likeness. Collectively, these findings position ERCO as a promising source of natural antioxidants and anti-diabetic agents, while identifying 2-epoxy-3,4-dihydroxycyclohexano[a]pyrene as a potential therapeutic lead. This investigation provides a foundation for future drug development, and, further experimental validations, isolation of active compounds, and subsequent clinical studies are required to validate its safety and efficacy.

## Introduction

There is a growing demand for novel chemical entities in drug discovery to facilitate the advancement of newer protein-based molecular targets. Natural products are playing a crucial role in meeting this requirement by means of ongoing investigations through the world's biodiversity, most of which is yet undiscovered. The utilization of natural resources especially plant-derived substances for therapeutic purposes has existed since the inception of mankind and still actively engaging. Most often synthetic drug metabolites have undesirable side effects along with therapeutic benefits. As a result of functioning pharmacologically and physiologically inside living cells, drugs derived from natural sources may thus have less unwanted effects [1]. WHO estimates that traditional medicine practices provide basic healthcare in about 80% population of African and Asian countries [2]. Traditional medicines are therefore cornerstone of interest among researchers over the world for finding the leads with a view to treating several complicated metabolic disorders, e.g., diabetes.

Hyperglycemia refers to the higher amount of sugar in the blood and is considered as the hallmark of diabetes mellitus (DM) [3].Moreover, prolonged hyperglycemia exacerbates the diabetic conditions and may further accelerate the creation of reactive oxygen species (ROS). Extreme ROS causes oxidative stress (OS), which plays a pivotal role in the development of long-term diabetes complications. Moreover, ROS-induced elevated OS levels can damage cells that are involved in insulin signaling. Insulin is a peptide hormone secreted by the pancreatic β-cells, playing an important role in regulating glucose homeostasis. Therefore, this damage can disrupt insulin's capacity to control blood glucose levels, thereby contributing to insulin resistance, which is a key determinant in the development of type-2 diabetes. Furthermore, this impairment in the functions of insulin exacerbates diabetes control and thereby contributes in the progression of diabetic complications, e.g., heart disease, nerve damage, and kidney failure [4]. However, Antioxidants neutralize ROS, protect the body from OS as well as support enzymatic defenses, prevent cellular damage, and diseases associated with OS [5].

Conventional diabetic treatment strategies are primarily contingent on insulin therapy and oral antidiabetic drugs, e.g., biguanides, sulfonylureas, *α*-amylase and

$\alpha$-glucosidase inhibitors, etc., for type 1 and type 2 DM, respectively. For instance, sulfonylureas promote insulin secretion by binding to the sulfonylurea receptor (SUR1) on pancreatic $\beta$-cells. Meanwhile, α-amylase and α-glucosidase are key pancreatic enzymes involved in carbohydrate digestion, each playing distinct roles in breaking down complex carbohydrates into simpler sugars for absorption. Both of these two enzymes elevate postprandial glucose levels and contribute to diabetic complications. Although effective in reducing postprandial glucose levels and improving overall glycemic control, conventional α-amylase and α-glucosidase inhibitors are associated with numerous adverse effects, including hypoglycemia, weight gain, cardiovascular complications, renal impairment, liver dysfunction, and insulin resistance [6]. This underscores the need for safer, plant-based alternatives with multi-target potential.

Medicinal plants serve as an alternative source of antidiabetic agents. Bioactive compounds from medicinal plants such as phenolics, flavonoids, tannins, carotenoids, and anthocyanins are well known to reduce blood sugar levels in diabetic patients by inhibiting α-amylase and α-glucosidase enzymes [7,8]. In addition, triterpenoids enhance glucose uptake and promote insulin secretion, thereby improving diabetic conditions [9].

*Curculigo orchioides* Gaertn (Family: Hypoxidaceae), locally known as Talmuli, is a perennial herb, thrives in many of Asia's subtropical regions, including China, India, Bangladesh, Malaya, Japan, and Australia [10,11]. Since ancient times, the Chinese traditional medicine system has utilized the plant's rhizome as a tonic to sustain vitality, wellness, and nourish the hepatic and renal systems. In Ayurvedic medicine, *C. orchioides* is used in the healing of jaundice, asthma, urinary and skin illnesses, bladder and kidney infections, piles, asthma, diarrhea, colic, gonorrhea, and sexual dysfunction. The herb is significantly acknowledged as an aphrodisiac in both traditional Chinese and Ayurvedic medicine [12]. As of 2023, a total of 111 secondary metabolites have been identified in its rhizome extracts, including three proteins, two polysaccharides, three sterols, five fatty acids, twelve phenolic glucosides, four phenolic compounds, two acids, fourteen chlorophenolic glucosides, six terpenoids, and eight cyclodipeptides. Additionally, this valuable medicinal plant contains flavonoids, alkaloids, and chlorinated compounds [10].

Traditionally, rhizome of the plant has been reported to possess neuroprotective [13], anti-osteoporotic [14], hepatoprotective [15], sexual behaviour and estrogenic [16], immunostimulatory [17], antidiabetic [18], antihypertensive [19], anticancer [20], antioxidant [21], antibacterial [22], antiasthmatic [23] anti-inflammatory, antidiarrhoreal, and hypoglycemic activities [24]. Leaf extract of the plant has been found to have anti-parkinson [25], anti-proliferative [26], antidepressant, and cytotoxic properties [27]. Moreover, researchers reported the root extract to exhibit cytotoxic [28], neuroprotective [29], and hepatoprotective activities [30].

Another scientific investigation reported that the rhizome of the plant contains a diverse range of phytocompounds from various phytochemical groups, including alkaloids, fatty Acids, esters, sterols, glycosides, polysaccharides, and carotenoids. In contrast, leaves were found to possess various sulfur-containing compounds, steroids, fatty acids, esters, nitrogenous alkaloid-like, and fused heterocyclic compounds [31].

Although the rhizomes of *C. orchioides* have been extensively studied, its roots remain largely unexplored, with only a few reports available. Furthermore, despite the presence of promising bioactive phytoconstituents, no studies have specifically focused on the screening for potential antidiabetic lead compounds. In this investigation, we evaluated the antioxidant and antidiabetic properties of ethanolic extract of *C. orchioides* roots (ERCO) on Swiss albino mice, followed by GC-MS analysis as well as ADMET analysis to assess its druggability properties.

## Materials and methods

### Ethics statement

This study's experimental protocols and procedures were approved by the Ethics Committee for the Care and Use of Laboratory Animals at Khwaja Yunus Ali University, Bangladesh (Approval No. KYAU/DEAN/EGC/2024/015). Mice were administered orally with 2% glucose solution to stop hypoglycemia subsequent an alloxan injection. At the end of the experiment, mice were euthanized using sodium thiopental (> 100 mg/kg) administered intraperitoneally to lessen any distress or pain.

## Plant materials collection and extraction process

*C. orchioides* was collected from the local fields of Belkuchi, Sirajganj, Bangladesh, in November 2023. After removing unwanted materials, the roots were washed with water and allowed to dry in the shade for a week and then it was grounded into a coarse powder using a grinder (Capacitor start motor, Wuhu motor manufacture, China), and ethanol was used as a solvent in the Soxhlet extraction process. 450 gm of powder was macerated with 2.25 liters 95% of ethanol in an amber glass container for a period of 15 days with occasional stirring with a glass rod, maintaining a solvent-to-sample ratio of 5:1 (v/w). The ratio was selected in accordance with the standard practices in phytochemical research to ensure solvent infiltration with complete maceration and maximize the extraction of bioactive constituents. Finally, it was filtered and rotary evaporated at 50 °C to obtain crude extract.

## Phytochemical screening

The phytochemical screening experiments were performed on ERCO to identify phytochemicals according to the previously published method [32].

## GC-MS analysis for the identification of phytoconstitutents

An instrument, Clarus 690 Gas Chromatograph (PerkinElmer, CA, MA, USA) was used to detect the phytocompounds present in the ERCO extract as previously described [32]. Sample was introduced into chamber using helium and the flow rate was 1 ml/min. The Gas Chromatograph was operated with a mass range of 50–600 m/z and a fixed scan time of 1 second. Phytocompounds in the extract were identified by comparing their mass spectra with those in the National Institute of Standards and Technology (NIST) database [32].

## Evaluation of antioxidant property

**Determination of total phenolic content (TPC).** The TPC of ERCO was determined by the Folin-Ciocalteu (FC) method with slight modifications [33]. Briefly, 0.5 ml extract (1 mg/mL) was mixed with 5 mL Folin-Ciocalteu reagent (1:10 v/v in distilled water) and 4 ml of 7.5% sodium carbonate followed by vortex for 15 seconds. The absorbance was measured at 765 nm with a spectrophotometer. The TPC values were expressed in terms of mg of gallic acid equivalent (GAE) per gram of dry extract, which is a common reference compound [33].

**Determination of total flavonoid content (TFC).** A colorimetric test was used to determine the TFC of the ERCO. Briefly, 5 ml sample of extract (stock solution, 1 mg/ml) added with $AlCl_3$ reagent followed by incubation at room temperature for 40 minutes. The absorbance was measured at 510 nm. The total flavonoid was calculated as a percentage of quercetin equivalents (% QE) dry weight [33].

**Determination of total tannin content (TTC).** The FC method was used to determine the TTC of ERCO [34]. 0.1 mL of the sample extract was added with 7.5 mL of distilled water, 0.5 mL of FC reagent, and 1 mL of 35% sodium carbonate solution. The mixture was kept at room temperature for 30 min. Then absorbance was measured at 725 nm.

**Ferric reducing antioxidant power (FRAP) assay.** The FRAP assay was performed according to Zhong *et. al*, with some modifications [35]. 1mL of extract or ascorbic acid (as standard) was mixed with 2.5 mL of phosphate buffer (200 mM; pH 6.6), and 2.5 mL of potassium ferricyanide (1%) allowed to incubate at 50ºC for 20 min. After that, added 2.5 mL of 10% TCA in the sample and centrifuged (3000 rpm for 10 min). Finally, 2.5 mL supernatant mixed with 2.5 mL of distilled water and 0.5 mL of $FeCl_3$ (0.1%). The absorbance was measured spectrophotometrically at 700 nm.

## Experimental animals

In this study, Swiss White Albino mice of both sexes (3–4 weeks old) were used and weighed 23–27 g. The mice were purchased from the Department of Pharmacy, Jahangirnagar University, Dhaka, Bangladesh. The mice were housed at a temperature of 25°C with 12 hours/12 hours darkness photoperiod and fed on rodent pellets and water ad libitum.

## Chemicals and reagents

Quercetin, gallic acid, aluminum chloride, chloroform, potassium ferricyanide, dibasic phosphate, DPPH, DMSO, ethanol, ferric chloride, ascorbic acid, methanol, n-hexane, sodium carbonate, sodium monobasic phosphate etc. were of analytical grade and were purchased from Mark, Germany. Folin–Ciocalteu (FC), *α*-amylase, *α*-glucosidase, alloxan, *p*-nitrophenyl-*α*-D-glucopyranoside (pNPG), were collected from Sigma ltd., St. Louis, USA. Acarbose and glibenclamide were collected from Pacific Pharma, Bangladesh.

## Acute toxicity study

Acute toxicity test of ERCO was carried out on mice to assess the toxicity of the extract. The experiment included test groups and control group, each comprise five mice. Five different doses of extract such as 50, 100, 500, 1000, and 2000 mg/kg of ERCO were administered orally to the test group following fasting overnight whereas the control group was administered 10% Dimethyl Sulfoxide (DMSO) solution. After the administration, the mice were monitored for 48 hours if there were any mortality or behavioral changes like tremor, morbidity, convulsion, writhing reflex, etc. [36].

## Assessment of anti-diabetic activity

**Determination of *α*-amylase inhibitory activity by extract.** *α*-amylase inhibitory activity of ERCO was determined by starch-iodine test via spectrophotometry at 610 nm [37]. In this assay, sample (1 ml), 20 µL of *α*-amylase solution (2 units/mL), and 1 mL of phosphate buffer were combined and incubated at 37°C for 10 minutes. After incubation, starch solution added and further incubated for 1 hour. Additionally, 200 µL of 1% iodine solution (comprising 5 mM I2 and 5 mM KI) was included followed by the addition of 10 mL of distilled water to each tube.

*α*-amylase inhibition by extract was determined by:

$$\% \ inhibition \ activity \ = \frac{Absorbance \ of \ sample - Blank \ Absorbance \ (Without \ enzyme)}{Blank \ Absorbance \ (Without \ enzyme) - Control \ Absorbance} \times 100$$

**Determination of *α*-glucosidase inhibitory assay.** The capability of the extract to suppress the activity of the *α*-glucosidase was assessed by established protocol via spectrophotometry at 405 nm [37]. To perform this test, 112 µL PBS (pH 6.8), *α*-glucosidase solution (20 µL, 1 unit/ mL), and 10 µL of the sample was taken together. After the reaction (15 min, 37°C), 20 µL of pNPG (2.5 mmol/L) was added to the mixer. Acarbose was considered as *α*-Glucosidase inhibitor in this assay.

The inhibitory activity was calculated using this formula:

$$\% \ inhibition \ of \ enzyme \ = \frac{Sample \ absorbance - sample \ blank \ absorbance}{Control \ absorbanc - blank \ absorbance} \times 100$$

**Induction of hyperglycemia.** Hyperglycemia on mice was developed by a single dose intraperitoneal (IP) administration of 150 mg/kg body weight of a newly prepared 10% alloxan monohydrate. After 2 days, blood glucose level (BGL) above 200 mg/dL on mice was considered diabetic. Prior to the experiment, the animals were fasted for 8–12 hours, with free access to water until the end of this test [38].

**Experimental design.** For either IP or oral drug administration, the diabetic mice were randomly divided into 5 groups (6 mice in each group). Group I consisted of normal mice orally administered with 0.1 mL physiological saline; group II consisted of diabetic mice (AIDM); group III was given 0.025 insulin units IP (0.25 insulin units in 1 mL) (1 IU/kg body weight) in 0.1 mL physiological saline. Group IV and V were given ERCO orally at a dose of 100 mg/Kg and 200 mg/kg body weight of the extract.

**Measurement of glucose level.** ERCO and the insulin were administered daily for two weeks and BGL was determined two hours after the last dose using Glucometer (VivaChek Biotech (Hangzhou) Co., Ltd, China).

## In silico study

### Phytochemicals selection and preparation

For the current inquiry, phytochemicals of *C. orchioides* were retrieved from the GCMS analysis. The list of these phytochemicals is presented in the S1 Table in the Supplementary file. The 3D conformers of ligands in SDF were obtained using the PubChem (3D conformer) database [39]. Using the Open Babel GUI [40], we next created a ligand library from this data that included every phytochemical that has been found. With PyRx 0.8 and Open Babel 2.3.1, the ligand library's phytochemicals were optimized, and their energy was minimized using the MMFF94 force field [41]. Afterward, AutoDock Tools was executed to convert all ligand files to.pdbqt files.

### Protein selection and preparation

When selecting a protein for molecular docking in diabetes we have chosen those proteins involved in glucose metabolism (e.g., *α*-amylase, *α*-glucosidase, and SUR1) or oxidative stress regulation (e.g., GPx, CAT and human peroxiredoxin).

Several studies found that α-amylase [42] was significantly higher in type 2 diabetes; reduced catalase (CAT) levels might be a consequence of chronic hyperglycemia and oxidative stress [43]; α-glucosidase can improve glycemic control and may also reduce insulin resistance [44]; decrease in Glutathione peroxidase (GPx) activity is thought to make the body less capable of neutralizing oxidative stress [45]; human peroxiredoxins play a crucial role in reducing oxidative stress in diabetic patient [46]; and mutations in the SUR1 gene could result in impaired insulin secretion [47] suggesting their potential use as a marker for metabolic control in diabetes. The 3D crystallographic structure of the reported proteins including **α**-amylase (PDB:1hny), CAT (PDB:2cag), α-Glucosidase (PDB:3top), GPx (PDB:2p31), human peroxiredoxin (PDB:1hd2), and SUR1 (PDB:5yw7) were retrieved from the Research Collaboratory for Structural Bioinformatics Protein Data Bank (RCSB PDB) (https://www.rcsb.org/) [48]. Hydrogen, ligands, and heteroatoms were removed from the complexes with the use of the BIOVIA Discovery Studio Visualizer during the processing step. The Swiss PDB Viewer software was utilized to incorporate the absent amino acid residues. Subsequently, hydrogen atoms were introduced to the amino acid residues [49]. Furthermore, the Swiss PDB Viewer software is utilized to minimize protein energy by placing the proteins in vacuo with the 43B1 force field. After that, the proteins were saved in.pdb format and converted into.pdbqt file formats.

### Molecular docking

Specific protein-ligand docking was carried out using Autodock Vina, which was integrated into PyRx 8.0, to predict the binding interactions of screened phytochemicals with the active sites of protein receptor targets [41]. Autodock Vina is a docking software that allows greater ligand flexibility, resulting in more accurate docking results [50]. The specific docking method is done by setting the grid box according to the active site that we previously identified using BIOVIA Discovery Studio Visualizer 2020 20.1.0 software. Analysis of binding sites and chemical bonds formed between proteins and ligands was carried out using PyMOL and Biovia Discovery Studio 2020 20.1.0 software [51].

### Drug likeness and in silico pharmacokinetics prediction

The free web tool SwissADME accessible at www.swissadme.ch, developed by the Swiss Institute of Bioinformatics, was used to computationally assess ADME analysis and drug-likeness [52]. This screening process was specifically conducted on the compounds that had high binding energy scores. Basic physicochemical parameters were calculated, including the Rule of 5 proposed by Lipinski, which is used to assess drug-likeness, primarily focusing on a molecule's oral absorption [53]. In addition to Lipinski's rule are also utilized to determine drug similarity possibilities [54]. Additionally, SwissADME was also used to conduct in-silico toxicity studies.

## Molecular dynamics simulation analysis

The Desmond software, created by Schrödinger LLC, was used to run the molecular dynamics simulation for a whole hundred nanoseconds [55]. Protein and ligand docking was performed prior to MD simulations in order to establish the static binding position of the molecule at the protein active site [56]. To predict the physiological state of ligand binding, MD simulations frequently employ Newton's classical equation of motion [57]. This allows them to simulate the movements of atoms throughout time. With the help of Maestro's Protein Preparation Wizard, the ligand-receptor complex was subjected to preprocessing. This process included optimization, minimization, and, if necessary, the addition of any residues that were lacking. In addition, the system was constructed with the help of the program known as System Builder. The TIP3P solvent model, which is based on an orthorhombic box, was applied at a temperature of 300 K, a pressure of 1 atm, and with the OPLS_2005 force field [58]. To replicate physiological conditions, the models were neutralized with counter ions and 0.15 M sodium chloride. The models were de-emphasized before running the simulation, and the resulting paths were recorded for analysis at 100 ps intervals.

## Statistical analysis

Data were calculated and presented as mean ± SEM, with each group consisting of 6 mice. Analysis of the data was performed using un-paired $t$-test. The figures were drawn using GraphPad Prism V8.0.2.

# Results

## Phytochemical analysis

The phytochemical screening of the ERCO extract confirmed the presence of various bioactive compounds, including carbohydrates, alkaloids, tannins, flavonoids, saponins, steroids, and quinones (Table 1).

## GCMS investigations

The investigation showed twenty-three (23) phytocompounds present in the extract as presented in S1 Table. Among 23 compounds, hexadecamethyl was showed highest peak area (29.82%) in the GCMS chromatogram (S1 Fig).

## Evaluation of antioxidant property

Secondary metabolites TPC, TFC, and TTC in ERCO extract were found to be 44.055 mg GAE/gm, 0.6768 mg QE/gm, and 103.375 mg TAE/gm of dry weight extract, respectively. The amount of TPC and TFC were calculated using

**Table 1. Phytochemical groups present in ERCO.**

| Phytochemical test | Results |
|---|---|
| Carbohydrate | + |
| Alkaloids | + |
| Tannins | + |
| Flavonoids | + |
| Saponin | + |
| Gum | − |
| Terpenoids | − |
| Glycosides | − |
| Quinones | + |
| Protein | − |
| Steroid | + |

the equation shown in S2 Fig. The calibration curves for gallic acid, quercetin, and tannic acid showed high linearity, indicating accurate and reliable quantification of phenolic and flavonoid compounds in ERCO. The gallic acid standard demonstrated a good linear fit with an absorbance range of 0.08–0.19 across concentrations of 0.25–5 µg/mL. Quercetin exhibited an absorbance range of 0.15–0.5 for concentrations of 125–500 µg/mL, reflecting its higher sensitivity. Tannic acid displayed absorbance values from 0.07–0.13 at concentrations of 20–100 µg/mL. These findings support the extract's significant bioactive compound content, which may contribute to its antioxidant and anti-diabetic activities.

### Ferric reducing antioxidant power (FRAP) assay

The ERCO extract exhibited strong ferric reducing power, which increased progressively with concentration up to 500 µg/mL. Ascorbic acid, used as the standard, displayed a concentration-dependent enhancement in reducing power, similarly, the reducing power of ERCO also demonstrated a concentration-dependent trend, depicted in Fig 1. At the maximum concentration (500 µg/mL), both the ERCO extract and ascorbic acid reached their highest absorbance values, indicating potent antioxidant activity. These results suggest that the antioxidant capacity of ERCO is positively correlated with its concentration, highlighting its potential as a natural antioxidant source.

### Acute toxicity study

The ERCO did not cause any mortality in the mice throughout 48 hours. Additionally, there was no sign of toxicity or abnormalities throughout the 7-day observational period, even at 3000 mg/kg body weight.

### Determination of *α*-amylase inhibitory activity

The ERCO exhibited a concentration-dependent inhibition of α-amylase, as shown in Fig 2A. The $IC_{50}$ value for the extract was determined to be 84.17 µg/mL, compared to 21.34 µg/mL for the standard drug acarbose. Although the extract demonstrated a slightly weaker inhibitory effect than acarbose, its inhibition curve suggests notable enzyme inhibition potential.

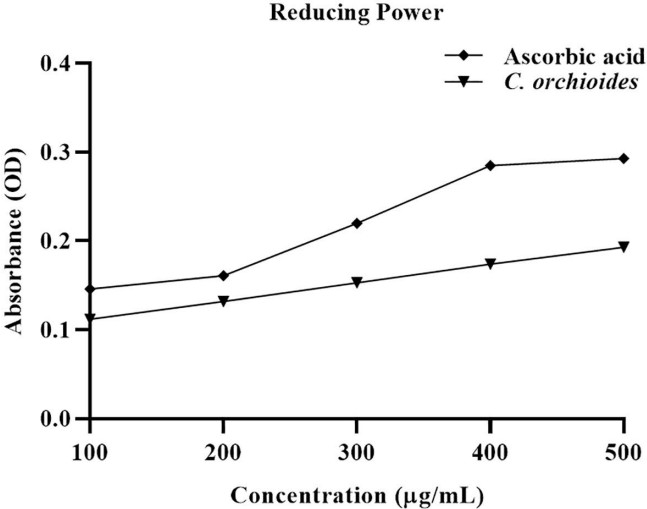

**Fig 1. Concentration vs. absorbance of Ascorbic acid and C. orchioides for reducing power determination.**

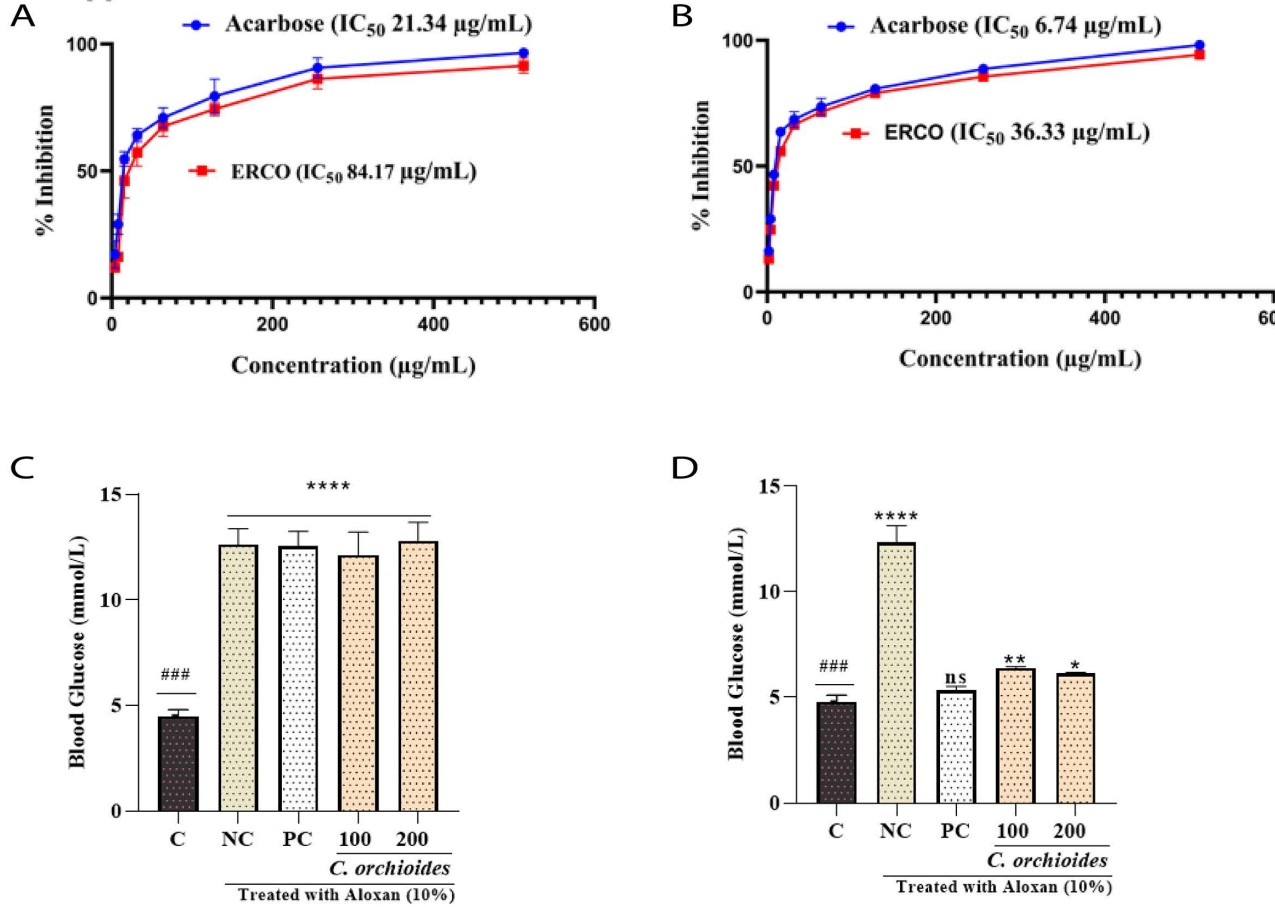

**Fig 2. (A)** Determination of α-amylase inhibitory activity by ERCO extract compared to the standard acarbose **(B)** Determination of α-glucosidase inhibitory activity by ERCO extract compared to the standard acarbose. C & D indicates the effects of ERCO extract to decrease BGL in normal and AIDM after two weeks of treatment **(C)** Blood glucose level before two weeks of treatment (on 13th day) on mice **(D)** Blood glucose level after two weeks of treatment (on 15th day).

### Determination of *α*-glucosidase inhibitory assay

Similarly, the ERCO extract displayed significant inhibition of *α*-glucosidase (Fig 2B), with an $IC_{50}$ value of 36.33 μg/mL compared to 6.74 μg/mL for acarbose. These results indicate that the extract effectively inhibits *α*-glucosidase activity, albeit less potent than the standard. In a nutshel, the results demonstrate that ERCO possesses promising inhibitory effects on both enzymes, supporting its potential use in managing postprandial hyperglycemia.

### Assessment of *in-vivo* anti-diabetic activity

Oral treatment with ERCO reduced BGL at all doses, as seen in Fig 2C, 2D and Table 2, as compared to the control group. From the first to the twenty-fourth hour, the BGL in the diabetic group increased gradually. On the other hand, insulin (as positive control) reduced the BGL 57.48% compared to the AIDM group. Moreover, the ethanol extract of *C. orchioides* reduced the BGL of 47.28% and 52.11% at the concentration of 100 mg/kg and 200 mg/kg BW respectively. The data confirms the hypoglycemic potential of ERCO, supporting its potential as an anti-diabetic agent.

**Table 2. Data for the blood glucose level (BGL) in normal and alloxan-induced diabetic mice (AIDM) after two weeks of treatment.**

| Treatment | Before Two Weeks (On 13th Day) | After Two Weeks (On 15th Day) | %Reduction of blood glucose level |
|---|---|---|---|
| Normal | 4.5±0.3082 | 4.8±0.3082 | – |
| AIDM | 12.64±0.737 | 12.36±0.7229 | – |
| AIDM + Insulin | 12.56±0.6989 | 5.34±0.1964 | 57.48% |
| AIDM + *C. orchioides* (100 mg/kg BW) | 12.14±1.096 | 6.4±0.0632 | 47.28% |
| AIDM + *C. orchioides* (200 mg/kg BW) | 12.82±0.8679 | 6.14±0.051 | 52.11% |

### *In silico* study

#### Molecular docking and protein-ligand interactions analysis

The docking score of the 20 phytochemicals with corresponding control ligands are presented in S2 Table in the supplementary file. Based on the binding affinity, selected only two phytochemicals including 1, 2-epoxy-3,4-dihydroxycyclohexano[a]pyrene and Anthracene (Fig 3) were then studied for drug-likeness and *in silico* Pharmacokinetics ADMET (absorption, distribution, metabolism, excretion, and toxicity) properties analysis. Acarbose was used as a reference drug for molecular docking and dynamics simulation studies to compare our findings.

The compound 1, 2-epoxy-3,4-dihydroxycyclohexano[a]pyrene exhibited strong interactions with multiple target enzymes, consistently showing higher binding affinities compared to standard inhibitors. For α-amylase (1HNY), the compound demonstrated a binding affinity of −9.9 kcal/mol, significantly stronger than acarbose's −7.6 kcal/mol. It formed Pi-Pi stacking interactions with TRP59 and Pi-alkyl interactions with LEU165, contributing to stable binding in the enzyme's active site. Acarbose, on the other hand, formed several hydrogen bonds and Pi-alkyl interactions with residues such as ARG421 and TRP280, but with a lower overall binding affinity [Fig 4A, B]. Similarly, for α-glucosidase (3TOP), 1,2-epoxy-3,4-dihydroxycyclohexano[a]pyrene exhibited a stronger binding affinity of −8.8 kcal/mol compared to Acarbose's −8.3 kcal/mol. This interaction was stabilized by Pi-Pi stacking with PHE1358 and Pi-alkyl interactions with PRO:1359 and VAL:1361 [Fig 4C, D].

For SUR1 (5YW7), the compound displayed a slightly stronger binding affinity (−9.1 kcal/mol) compared to the standard inhibitor Glibenclamide (−9.0 kcal/mol), forming Pi-Pi stacking interactions with TYR377 and Pi-alkyl interactions with LEU434 [Fig 4E, F]. This indicates that the compound could act as a more effective inhibitor in regulating the enzyme's

1, 2-epoxy-3,4-ihydroxycyclohexano[a]pyrene
(CID: 41322)

Anthracene
(CID: 818)

**Fig 3. Structures of compounds identified through GCMS analysis that showed highest binding affinities against human glutathione peroxidase, peroxiredoxin 5, Catalase, sulfonylurea receptor 1 (SUR1), *a*-amylase, and *a*-glucosidase.**

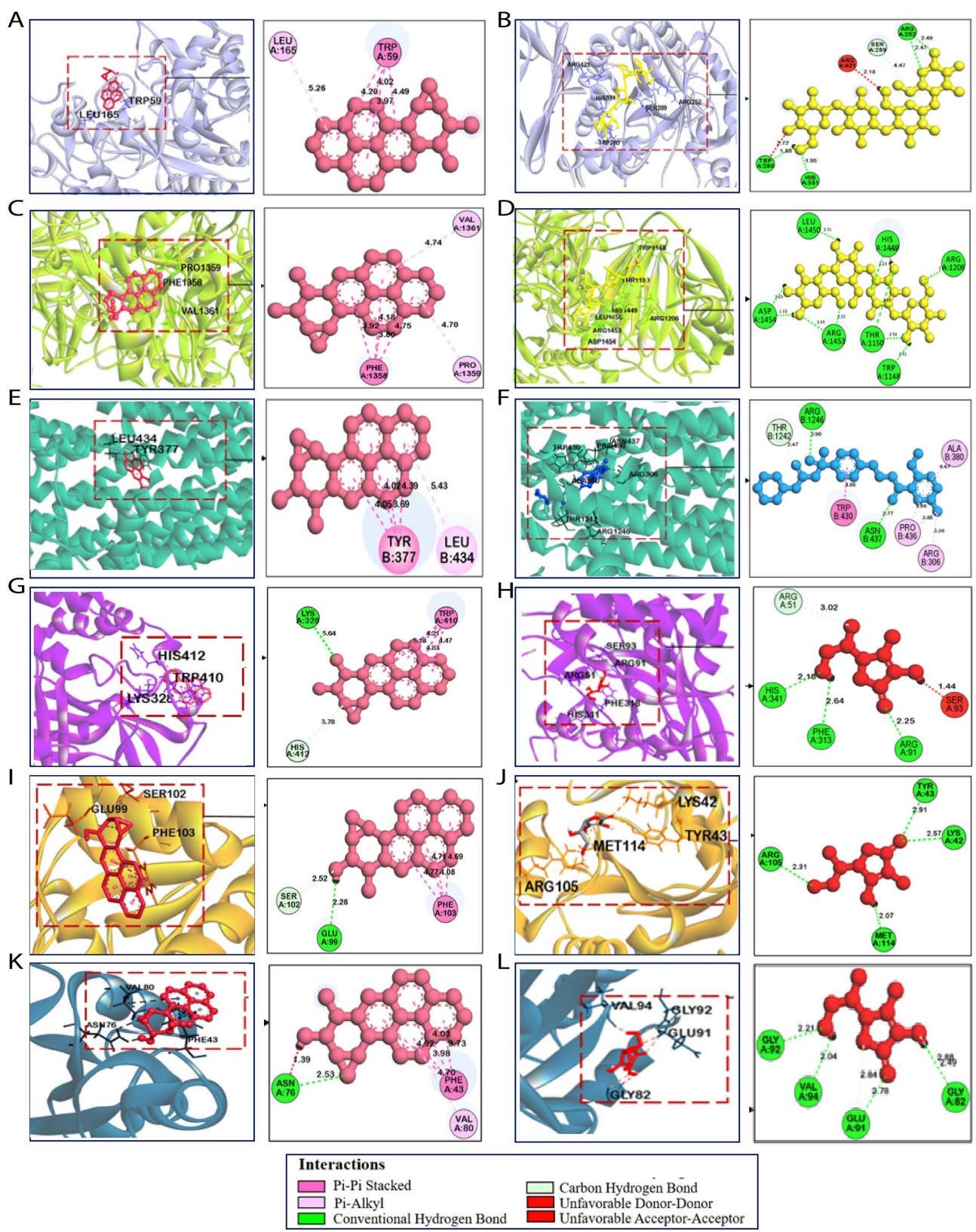

**Fig 4. In the plot, the above panel shows the 2D structure of corresponding compounds.** 3D and 2D molecular docking interactions of (A) CID: 41322 1, 2-epoxy-3,4-dihydroxycyclohexano[a]pyrene (Pink) and (B) Acarbose as a standard (Yellow) in the active site of α-amylase (PDB ID: 1HNY);

(C) CID: 41322 1, 2-epoxy-3,4-dihydroxycyclohexano[a]pyrene (Pink) and (D) Acarbose as a standard (Yellow) in the active site of α-Glucosidase(PDB ID: 3TOP); (E) CID: 41322 (Pink) and (F) Glibenclamide as a standard (Blue) in the active site of SUR1 (PDB ID: 5YW7); (G) CID: 41322 (Pink) and (H) Ascorbic acid as a standard (Red) in the active site of Catalase (PDB ID: 2CAG); (I) CID: 41322 (Pink) and (J) Ascorbic acid as a standard (Red) in the active site of Glutathione Peroxidase (PDB ID: 2P31); (K) CID: 41322 (Pink) and (L) Ascorbic acid as a standard (Red) in the active site of Peroxiredoxin 5 (PDB ID: 1HD2).

activity. In the case of Catalase (2CAG), 1,2-epoxy-3,4-dihydroxycyclohexano[a]pyrene exhibited a significantly stronger binding affinity (−8.2 kcal/mol) compared to Ascorbic Acid (−5.5 kcal/mol). The ligand's stable binding was achieved through Pi-Pi stacking with TRP410 and conventional hydrogen bonds with HIS412 and LYS328 [Fig 4G, H]. With Glutathione Peroxidase (2P31), the compound also demonstrated superior binding affinity (−7.0 kcal/mol) compared to Ascorbic Acid (−5.4 kcal/mol), interacting with GLU99, SER102, and PHE103 through hydrogen bonds and Pi-Pi stacking. This stable interaction supports its potential as a potent inhibitor of glutathione peroxidase [Fig 4I, J]. Finally, in Peroxiredoxin 5 (1HD2), 1,2-epoxy-3,4-dihydroxycyclohexano[a]pyrene displayed a binding affinity of −7.6 kcal/mol, outperforming ascorbic acid (−5.2 kcal/mol). The compound formed Pi-Pi interactions with PHE43 and hydrogen bonds with ASN76 [Fig 4K, L], further enhancing its stable binding in the enzyme's active site. The interactions of the amino acid residues are listed in Table 3.

### ADMET and drug likeness properties analysis

By examining key parameters like absorption, distribution, metabolism, excretion, and toxicity, we can identify drug candidates with the most promising profiles for further study. The ADMET analysis (S3 Table) of 2-epoxy-34-dihydroxycyclohexano[a]pyrene (41322) and Anthracene (8418) indicates that both compounds exhibit high intestine absorption and low skin permeability, rendering them appropriate for oral administration. Intestinal absorption, skin permeability, and Caco-2 permeability are widely used to predict drug absorption and oral bioavailability. Anthracene exhibits superior distribution and clearance rates, whereas compound 41322 demonstrates slightly enhanced permeability across the blood-brain barrier. Both are substrates of CYP3A4 and do not inhibit significant metabolic enzymes, hence minimizing drug interaction risks. In terms of toxicity, both compounds tested positive for mutagenicity (AMES test), with Anthracene posing a risk of skin sensitization. Although both have moderate oral bioavailability, Anthracene violates one of Lipinski's rules. Anthracene (8418) tested positive for mutagenicity (AMES test), showed skin sensitization, and violated one of Lipinski's rules, whereas 2-epoxy-34-dihydroxycyclohexano[a]pyrene (41322) only showed mutagenicity and complied fully with Lipinski's Rule of Five. Therefore, we selected 2-epoxy-34-dihydroxycyclohexano[a]pyrene (41322) for molecular dynamics simulation and further studies. The key pharmacokinetic properties of compound 41322 are illustrated in the bioavailability radar plot shown in Fig 5.

### Molecular dynamics simulation studies

The simulations were carried out to evaluate the overall stability and flexibility of the ligand-protein complexes of CID: 41322 against *α*-amylase (PDB: 1hny), CAT (PDB:2cag), α-Glucosidase (PDB:3top), GPx (PDB:2p31), human peroxiredoxin (PDB:1hd2), and SUR1 (PDB:5yw7) proteins.

### Root mean square deviation (RMSD) analysis

RMSD (Root Mean Square Deviation) is used to assess the stability and structural deviation of protein-ligand complexes over time. Throughout the simulation, RMSD computes the average distance between atoms in a given structure and those in subsequent structures. The RMSD analysis (Fig 6A) reveals that CID:41322 exhibits significantly better stability with the 1hny protein, with an average RMSD of 4.845 Å, compared to the control (Acarbase), which shows considerable instability with an average RMSD of 36.189 Å. However, for the 3top protein, Acarbase demonstrates greater stability

**Table 3. Binding affinities of compounds from ethanol extract of *C. orchhioides* against anti-diabetic and antioxidant receptors.**

| Protein (PDB ID) | Compound ID | Docking Score (k/mol) | Hydrogen Bonds | Other Bonds | Grid box center | Dimension size |
|---|---|---|---|---|---|---|
| α-amylase (1HNY) | 1, 2-epoxy-3,4-dihydroxycyclohex-ano[a]pyrene (41322) | −9.1 | – | TRP59, LEU165 | x = 8.32 y = 58.13 z = 22.75 | x = 66.28 y = 86.81 z = 58.50 |
| | Anthracene (8418) | −7.2 | – | TRP59, TYR62 | | |
| | Acarbose (41774_Standard) | −7.6 | ARG252, TRP280, HIS331 | SER289, ARG421 | | |
| α-Glucosidase (3TOP) | 1, 2-epoxy-3,4-dihydroxycyclohex-ano[a]pyrene (41322) | −8.8 | – | PHE1358, PRO1359, VAL1361 | x = −46.13 y = 21.74 z = 17.84 | x = 93.85 y = 63.40 z = 85.89 |
| | Anthracene (8418) | −9 | – | TYR1251, PHE1559, PHE1560, HIS1584 | | |
| | Acarbose (41774_Standard) | −8.3 | TRP1148, THR1150, ARG1206, HIS1449, LEU1450, ARG1453, ASP1454 | – | | |
| SUR1 (5YW7) | 1, 2-epoxy-3,4-dihydroxycyclohex-ano[a]pyrene (41322) | −9.1 | – | TYR377, LEU434 | x = 215.81 y = 218.43 z = 165.85 | x = 99.07 y = 62.61 z = 133.75 |
| | Anthracene (8418) | −7.8 | – | ALA380, TRP430, LEU434, PRO436 | | |
| | Glibenclamide (348_standard) | −9 | ASN437, ARG1246 | ARG306, ALA380, TRP430, PRO436, THR1242 | | |
| Catalase (2CAG) | 1, 2-epoxy-3,4-dihydroxycyclohex-ano[a]pyrene (41322) | −8.2 | LYS328 | TRP410, HIS412 | x = 60.83 y = 16.22 z = 4.76 | x = 68.79 y = 80.62 z = 94.31 |
| | Anthracene (8418) | −9.9 | – | PHE132, PHE140, HIS197, MET329, ARG333 | | |
| | Ascorbic acid (54670067_standard) | −5.5 | ARG91, PHE313, HIS341 | ARG51 | | |
| GPx (2P31) | 1, 2-epoxy-3,4-dihydroxycyclohex-ano[a]pyrene (41322) | −7 | GLU99 | SER102, OHE103 | x = −7.74 y = −1.70 z = −42.74 | x = 37.55 y = 31.83 z = 43.25 |
| | Anthracene (8418) | −5.8 | | PHE28, TYR131 | | |
| | Ascorbic acid (54670067_standard) | −5.4 | LYS42, TYR43, ARG105, MET114 | – | | |
| Peroxiredoxin 5 (1HD2) | 1, 2-epoxy-3,4-dihydroxycyclohex-ano[a]pyrene (41322) | −7.6 | ASN76, | PHE43, VAL80 | x = 8.98 y = 43.64 z = 19.04 | x = 40.99 y = 38.57 z = 48.64 |
| | Anthracene (8418) | −5.9 | – | GLU16, ALA90 | | |
| | Ascorbic acid (54670067_standard) | −5.2 | GLY82, GLU91, GLY92, VAL94 | – | | |

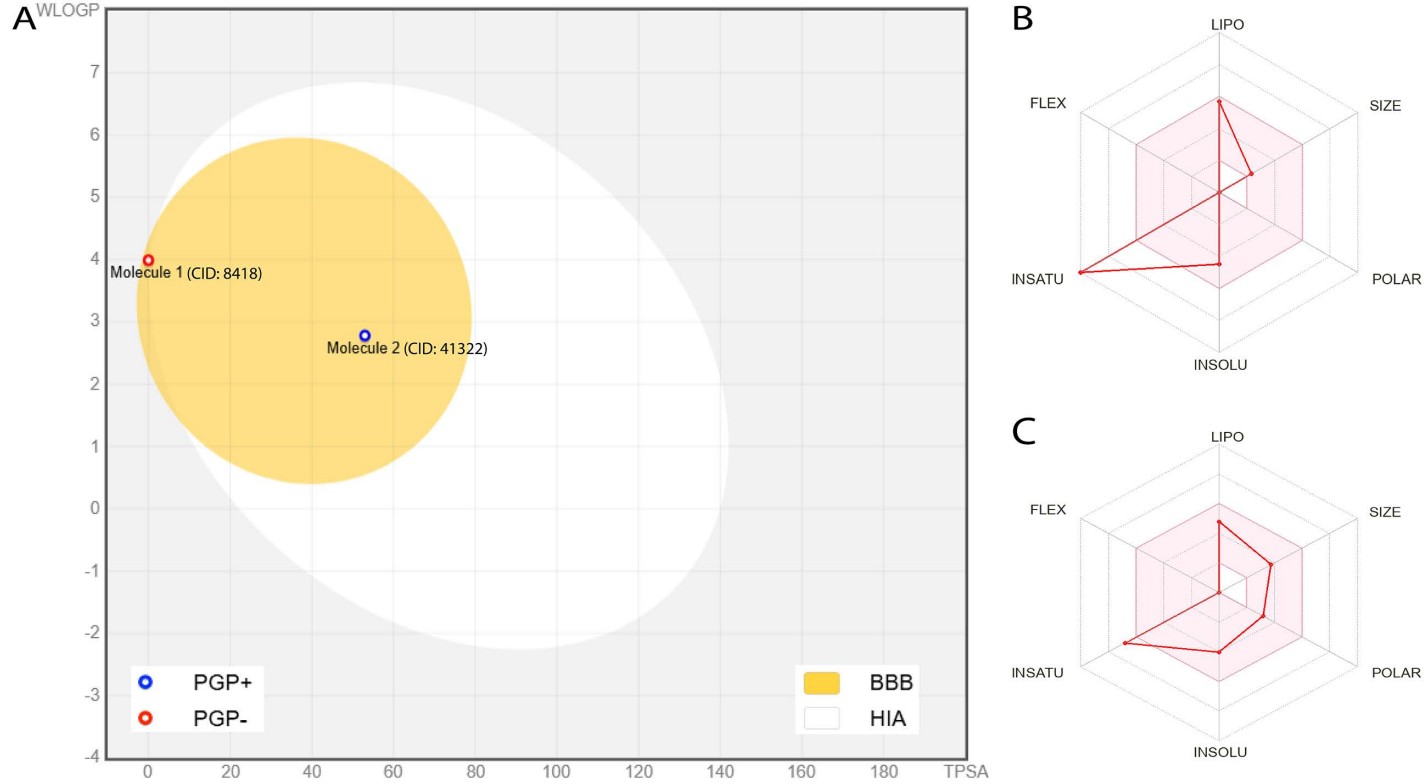

**Fig 5.** (A) BOILED-Egg model of Molecule 1 (CID: 8418) and Molecule 2 (CID: 41322) depicting blood–brain barrier (BBB, yellow region) and high gastrointestinal absorption (HIA, white region) predictions. Blue circle indicates P-glycoprotein substrate (PGP+); red circle indicates non-substrate (PGP−). (B, C) Bioavailability radar plots of CID: 8418 and CID: 41322, respectively, illustrating physicochemical descriptors: lipophilicity (LIPO), size (SIZE), polarity (POLAR), solubility (INSOLU), saturation (INSATU), and flexibility (FLEX).

with an average RMSD of 2.650 Å, while the 3top + CID: 41322 complex shows higher deviations with an average RMSD of 11.762 Å. Similarly, for the 5yw7 protein, the control (Glibenclamide) shows better stability with an average RMSD of 3.698 Å compared to the 5yw7+CID: 41322 complexes, which has a higher RMSD of 7.875 Å. Nonetheless, it's important to note that while the 5yw7+CID: 41322 complex initially exhibits more fluctuations, it eventually equilibrates and stabilizes by the end of the simulation. This suggests that despite the initial instability, CID: 41322 may still be effective in binding to the 5yw7 protein, potentially maintaining its therapeutic potential.

Similarly, The RMSD analysis (Fig 6B) reveals that for the 1HD2 protein, ascorbic acid (control) exhibits better stability with an RMSD of 5.161 Å, while CID: 41322 shows higher deviations with an RMSD of 23.146 Å. In contrast, for the 2cag protein, CID: 41322 provides excellent stability throughout the simulation with an RMSD of 4.844 Å, significantly outperforming ascorbic acid, which has a much higher RMSD of 26.381 Å. The 2p31 protein shows differing behavior: ascorbic acid (control) initially remains stable for the first 75 ns but then begins to fluctuate significantly, whereas the 2p31+CID: 41322 complex fluctuates between 20 ns and 120 ns before equilibrating and stabilizing for the remainder of the simulation.

## Root mean square fluctuation (RMSF) analysis

RMSF measures the flexibility of individual residues of the protein during the simulation. The RMSF analysis (Fig 6C) shows that, the 1hny protein, when bound to CID: 41322, displayed a lower average RMSF value of 1.075 Å compared

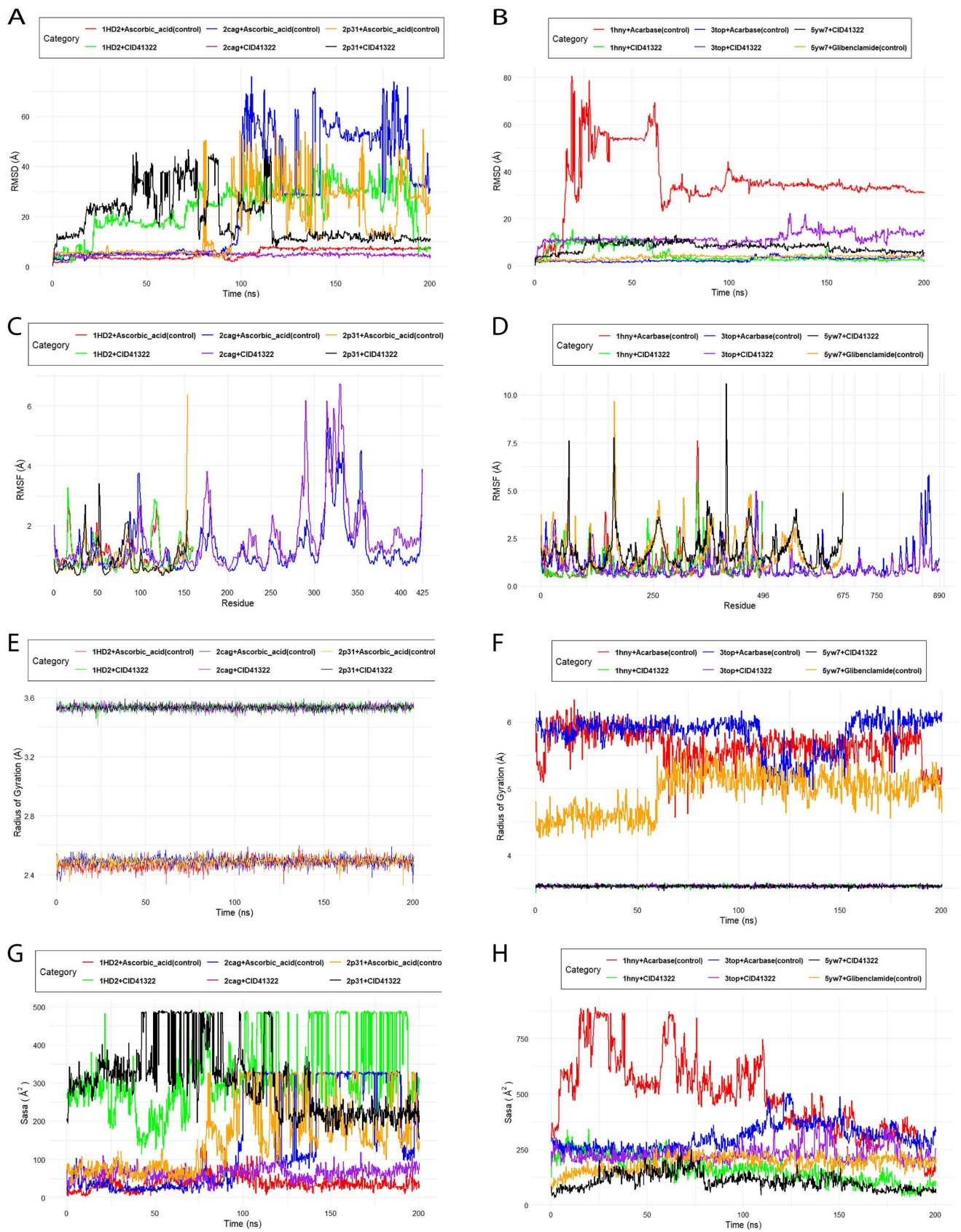

**Fig 6. Molecular dynamics trajectory including (A) RMSD, (C) RMSF, (E) Rg and (G) SASA analysis of two compounds CID: 41322 and control drug against the 1HD2, 2CAG, 2P31 protein structures for Antioxidant activity at the left panel. While in the right panel (B) RMSD, (D) RMSF, (F) Rg and (H) SASA analysis against the 1HNY, 5YW7 and 3TOP protein structures for Antidiabetic activity.**

to the control with Acarbase, which had an RMSF of 1.089 Å. In the CID: 41322 complex, GLY_351 showed a fluctuation of 5.497 Å, whereas the control with Acarbase exhibited a higher fluctuation of 7.262 Å at this residue. Similarly, the 3top protein showed enhanced stability with CID: 41322, reflected in a lower average RMSF of 0.9274 Å compared to 1.074 Å in the control, with SER_1440 experiencing the highest fluctuation at 4.996 Å. Conversely, the 5yw7 protein exhibited increased flexibility when complexed with CID: 41322, with an average RMSF of 1.9071 Å, higher than the control with Glibenclamide at 1.7262 Å, and ASP_1060 showing the highest fluctuation at 10.599 Å.

For the 1HD2 protein, CID:41322 shows slightly improved stability compared to the control with Ascorbic acid, with a lower average RMSF value of 0.9809 Å compared to 1.0451 Å. In contrast, for the 2cag protein, CID:41322 leads to greater flexibility, indicated by a higher average RMSF of 1.516 Å compared to 1.2405 Å for the control, particularly around residues 250–350, suggesting that CID:41322 induces more conformational changes. For the 2p31 protein, CID: 41322 maintains a stability profile similar to the control, with RMSF values of 0.84168 Å and 0.8135 Å for CID: 41322 and the control, respectively (Fig 6D).

### Radius of gyration analysis

The Radius of Gyration (Rg) analysis (Fig 6E) reveals that CID: 41322 significantly enhance the stability and compactness of the 1hny, 3top, and 5yw7 proteins compared to their respective controls. Specifically, the analysis shows two distinct clusters: one around 4.6 Å, where the CID: 41322 complexes for each protein (1hny, 3top and 5yw7) consistently exhibit lower Rg values, indicating a more compact and stable structure. In contrast, the control complexes (Acarbase for 1hny and 3top, and Glibenclamide for 5yw7) form an upper cluster with higher Rg values, fluctuating between 5.6–6.0 Å for 1hny and 3top, and stabilizing around 4.8 Å for 5yw7, reflecting greater flexibility and reduced stability in these protein structures.

The Radius of Gyration (Rg) analysis reveals that CID: 41322 effectively maintains the stability and compactness of the 1HD2, 2cag, and 2p31 proteins, although the Rg values of the control complexes indicate greater compactness. The plot shows that the Rg values for the CID: 41322 complexes with each protein are around 3.6 Å, while the control complexes of each protein are around 2.4 Å. Despite the higher Rg values observed with CID: 41322 for each protein, the complexes still maintain a stable structure throughout the simulation (Fig 6F).

### Solvent accessible surface area (SASA)

The SASA analysis (Fig 6G) indicates that CID: 41322 generally promote a more compact and stable protein structure across all three proteins compared to their respective controls. For the 1hny and 3top proteins, CID: 41322 significantly reduce solvent exposure, indicating a tighter, more stable conformation. The 5yw7 protein also shows a slight improvement in compactness with CID: 41322, maintaining lower SASA values compared to the control. For the 1HD2 and 2p31 proteins, CID: 41322 increases solvent exposure and reduces compactness compared to the control, as shown by the higher SASA values fluctuating between 200–450 Å². In contrast, for the 2cag protein, CID: 41322 maintains a similar level of compactness and stability to the control with Ascorbic acid, with SASA values remaining around 100–150 Å² (Fig 6H).

### Binding free energy analysis via MM/GBSA

As illustrated in Fig 7, the 1hny + Acarbase (control) shows a highly favorable total binding energy, driven by strong Coulombic and Van der Waals interactions, indicating robust binding. In contrast, the 1hny + CID: 41322 complex has less favorable total energy, suggesting weaker binding. Similarly, 3top + Acarbase (control) exhibits strong binding with

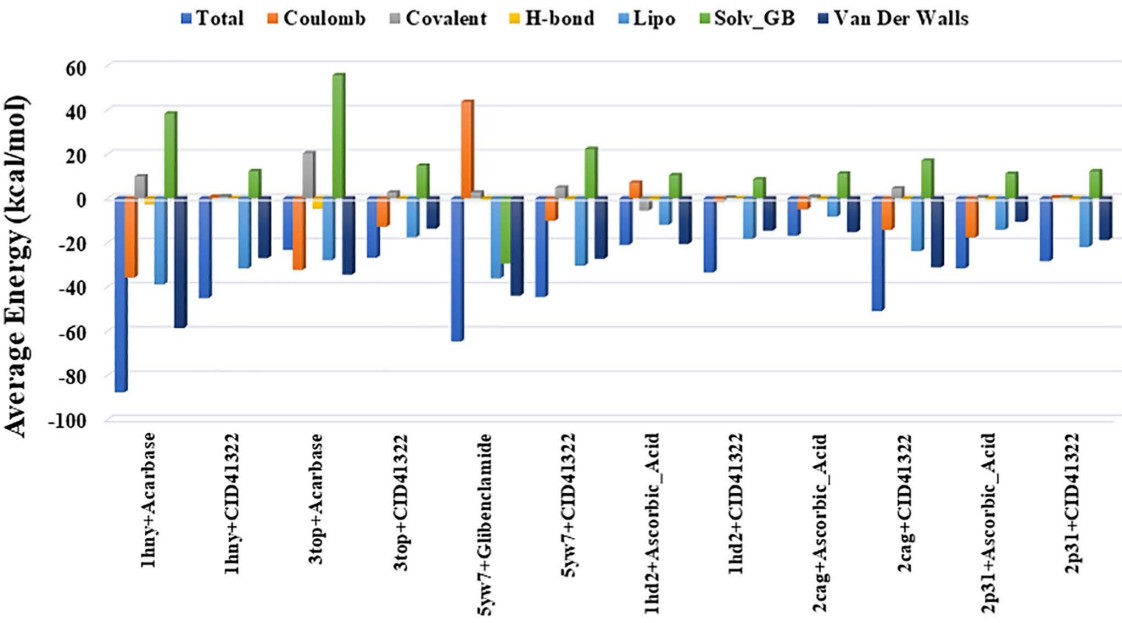

**Fig 7. Binding free energy calculation via MMGBSA analysis of reported proteins and compounds.**

substantial hydrogen bonding and Van der Waals contributions, while 3top + CID:41322 demonstrates reduced binding stability. For 5yw7 + Glibenclamide (control), the strong hydrophobic interactions result in a significantly negative total energy, whereas 5yw7 + CID: 41322 displays a less negative total energy, indicating weaker binding compared to the control.

For the antioxidant proteins, 1HD2 + Ascorbic Acid (control) shows stable binding with a more negative total energy due to favorable Coulombic and Van der Waals interactions. However, 1HD2 + CID: 41322 demonstrate a less favorable binding profile. In contrast, 2cag + CID: 41322 shows a more negative total binding energy compared to the control, suggesting that CID: 41322 may bind more stably to the 2cag protein. Finally, the 2p31 + CID: 41322 complex exhibits a binding energy profile comparable to the control, with hydrophobic and Van der Waals forces playing a significant role in maintaining stability.

## Discussion

Herbal medicines have long been valued for their therapeutic potential in managing various chronic diseases, including diabetes and oxidative stress-related disorders [59]. *C. orchiodes* is a traditional medicinal plant, rich in bioactive phyto-compounds such as flavonoids, saponins, and phenolic acids, which are known to exhibit potent antioxidant and antidiabetic activities [60]. Phytochemicals have defense mechanisms by nature and protect against various diseases. In our study, phytochemical analysis of *C. orchioides* revealed the presence of carbohydrates, alkaloids, tannins, flavonoids, saponins, steroids, quinones, and steroids (Table 1). These plant-based bioactive compounds serve as antioxidant and anti-hyperglycemic agents.

Antioxidants are recognized for their capacity to prevent disease by scavenging reactive species such as ROS and reactive nitrogen species (RNS), or by preventing cell oxidation. Antioxidants inhibit or postpone oxidative reactions, which in turn regulates the overproduction of oxidants [61]. Phenolic compounds were identified as major components of ERCO which are widely used to treat a range of ailments, such as carcinoma, low blood sugar, oxidative stress, wounds, skin

infections, and immune system dysfunction [11]. Comparable health benefits, including anti-cancer, anti-inflammatory, antiviral, anthelmintic, cardio-protective, neuroprotective properties, and antibacterial properties are provided by phenols and flavonoids [59]. Surveswaran et al. also demonstrated that the antioxidant activity of ERCO was validated using both the Ferric reducing ability of plasma (FRAP) assay [62]. Ratnam et al. reported that the ERCO showed remarkable reducing power, free radical scavenging, and antioxidant activities [63]. In our study, we found TPC value of 44.055 mg GAE/gm and TFC value of 0.6768 mg QE/gm, and TTC value of 103.375mgTAE/gm of dry weight extract from ethanol extract of *C. orchioides* roots. GC-MS analysis (S1 Table) and preliminary phytochemical analysis indicated that the ERCO contains some bioactive compounds that could contribute to the observed antioxidant activity.

The ERCO demonstrated significant anti-diabetic activity by reducing BGL by 47.28% and 52.11% at 100 mg/kg and 200 mg/kg BW, respectively, in AIDM, comparable to insulin's 57.48% reduction (Table 2, and Fig 3). These hypoglycemic effects may be attributed to the plant's bioactive compounds, such as flavonoids, phenolics, and tannins, known for enhancing insulin sensitivity and glucose uptake [64]. Additionally, ERCO showed potent inhibition of α-amylase and α-glucosidase enzymes, which delay carbohydrate digestion and help control postprandial hyperglycemia [65]. Singla K. and Singh R. reported that the ethanolic and hydroalcoholic extracts of *C. orchioides* effectively mitigated lipid alterations associated with hyperglycemia, oxidative stress, and renal dysfunction in streptozotocin-nicotinamide-induced diabetic nephropathy in rats [66]. Gulati *et al.* also found that the ethanolic extract of *C. orchioides* may suppress adipogenesis and promote glucose uptake in 3T3-L1 adipocytes [67]. The dual effects highlight the potential of *C. orchioides* as a natural anti-diabetic agent, warranting further studies for compound isolation and clinical validation.

Molecular docking, a structure-based drug design method, was utilized to assess the strength of binding and get insights into the potential interactions between ligands and proteins [55]. For this purpose, 20 phytochemicals of ERCO were docked against the reported six proteins. Its antidiabetic properties are primarily attributed to the modulation of insulin sensitivity and glucose metabolism [68]. Additionally, *Curculigo* exhibits antibacterial and anti-inflammatory effects, likely mediated by bioactive compounds such as curculigosides and phenolic acids [10]. After docking, based on the binding affinity we selected common two phytochemicals namely 1,2-epoxy-3,4-dihydroxycyclohexano[a]pyrene (binding affinity range between −7 to −9.1 kcal/mol) and anthracene (range between −5.8 to −9.9 kcal/mol) against all proteins for our further studies. Binding affinity refers to the degree of strength in the interaction, which can be used to predict whether an interaction between two molecules can form or not. The lower binding affinity value, the stronger and more stable the interaction between molecules [69]. In a nutshell, the best 1,2-epoxy-3,4-dihydroxycyclohexano[a]pyrene compound exhibited significant interactions with Lys328, Glu99 and Asn76 via hydrogen bond formation in catalase, GPx and peroxiredoxin 5 proteins, respectively. In drug design, hydrogen bonding interactions between drug molecules and the target can be studied in real-time, helping to optimize drug candidates before synthesis and experimental testing. Too many hydrogen bonds can make the drug overly polar, reducing its ability to cross cell membranes [70]. On the other hand, a vast number of residues including Trp59, Leu165, Phe1358, Pro1359, Val1361, Tyr1251, Phe1559 and Phe1560 etc. were formed via hydrophobic bond interaction in respective protein structures. In drug design, enhanced membrane permeability and robust target binding are two potential benefits of high hydrophobic interactions in drug design. The solubility, bioavailability, possibility of non-specific interactions, and toxicity associated with these interactions are further challenges [71].

ADMET profiling is essential in drug development as it evaluates the safety, efficacy, and pharmacokinetics of compounds, reducing the likelihood of late-stage failures [72]. Intestinal absorption, skin permeability, and Caco-2 permeability are widely used to predict drug absorption and oral bioavailability [73]. CID: 41322 demonstrates slightly higher intestinal absorption (96.08%) compared to CID: 8418 (95.51%) and better BBB permeability, suggesting potential for CNS activity. However, CID: 8418 shows higher Caco-2 permeability and tissue distribution (VDss), indicating broader systemic distribution. Toxicity plays a vital role by assessing the potential harmful effects of a compound, while drug-likeness evaluates its chemical properties to ensure it meets the criteria for development [54,74]. Toxicity analysis reveals no hepatotoxicity

for both compounds, though CID: 8418 shows potential for skin sensitization. Despite CID: 8418 violating one Lipinski rule due to its lipophilicity (Log P of 4.02), both compounds maintain a bioavailability score of 0.55, supporting drug-likeness. These properties indicate promising pharmacokinetics for further development [54].

The molecular dynamics simulations indicate variable stability and binding performance of CID: 41322 across different protein targets. RMSD helps evaluate how much the system changes from its initial structure, providing insights into the flexibility and binding stability of the complex. RMSD analysis revealed that CID:41322 exhibited better stability with α-amylase (PDB: 1hny), maintaining a lower average RMSD compared to the control, Acarbose complex [75,76]. RMSF measures the flexibility of individual residues of the protein during the simulation. Higher RMSF values indicate more flexible regions, while lower values suggest greater stability in those areas [77]. RMSF analysis revealed that CID: 41322 generally reduced flexibility in key residues of α-amylase and α-Glucosidase, contributing to greater structural stability, but increased fluctuations were observed for SUR1, suggesting more dynamic interactions.

The Radius of Gyration (Rg) is a measure of the compactness of a protein structure over time during the simulation. Lower Rg values indicate more compact and stable structures, while higher values suggest more flexibility or instability [78,79]. SASA measures the extent of the protein surface exposed to the solvent, providing insights into the compactness and stability of the protein-ligand complex. Lower SASA values indicate reduced solvent exposure, implying tighter packing and greater stability [80,81]. Radius of Gyration (Rg) and Solvent Accessible Surface Area (SASA) confirmed enhanced compactness for CID: 41322 in its complexes with α-amylase and α-Glucosidase, while antioxidant proteins like human peroxiredoxin (PDB: 1HD2) and GPx (PDB: 2p31) exhibited greater flexibility with CID: 41322 compared to controls. Interaction analysis highlighted stable hydrophobic and hydrogen bonding, particularly with LEU_328 in the 2cag complex, which may enhance binding stability. The MMGBSA analysis calculates the binding free energy between the protein and the ligand, providing an estimate of the binding affinity and stability of the complex. Lower binding energy values indicate stronger, more stable interactions [82,83]. Our study showed stronger binding of CID: 41322–2cag and comparable binding energies to controls in 2p31, supporting its potential for drug development. These findings indicate that CID: 41322 demonstrate selective binding and stability across various protein targets, warranting further experimental and *in silico* investigation for drug discovery.

Despite the promising findings, this study has several limitations that warrant consideration. First, although the antioxidant and anti-diabetic activities of ERCO were demonstrated, the underlying molecular mechanisms remain unclear and require further elucidation. The study primarily relied on in silico methods and in vitro analysis, which may not fully replicate in *invivo* conditions. Additionally, the bioavailability and metabolic fate of the identified bioactive compounds have not been investigated, which are critical for drug development. The sample size of tested compounds was also limited, potentially overlooking other significant bioactive molecules. Lastly, while ADMET profiling predicted favorable pharmacokinetics and toxicity profiles, experimental validation of these properties in animal models and clinical trials is essential to confirm drug-likeness and safety.

Despite its limitations, this study has some potential to capture the interest of researchers and provide valuable direction for those exploring drug discoveries from natural products.

## Conclusion

This study highlights the significant antioxidant and anti-diabetic potential of *C. orchioides* root extracts, attributed to its rich bioactive phytochemical content, including flavonoids and phenolic compounds. The ethanolic extract demonstrated potent reducing power, free radical scavenging, and enzyme inhibition activities, comparable to known reference standards. Additionally, molecular docking and dynamics simulations identified CID: 41322 as a promising candidate with strong binding stability across multiple protein targets, particularly α-amylase and *α*-Glucosidase. ADMET study projected favorable pharmacokinetic properties, supporting the compound's potential as a natural therapeutic agent. Despite the limitations, these findings suggest that *C. orchioides* could serve as a valuable resource for developing antioxidant and

anti-diabetic therapies. Further experimental validation, isolation of active compounds, and clinical studies are recommended to advance its development as a pharmaceutical agent.

## Supporting information

**S1 Fig. GCMS chromatogram of ERCO.**
(TIF)

**S2 Fig. Concentration vs. absorbance of Gallic acid for TPC determination.** (B) Concentration vs. absorbance of quercetin for TFC determination. (C) Concentration Vs absorbance of Tannic Acid for TTC determination.
(TIF)

**S1 Table. GCMS Analysis of ERCO.**
(DOCX)

**S2 Table. Docking Scores of 20 GC-MS-Identified Phytochemicals Compared with the Standard.**
(DOCX)

**S3 Table. ADMET and drug-likeness properties analysis of two compounds.**
(DOCX)

## Acknowledgments

The authors express their sincere gratitude to the authorities of Phytochemistry and Pharmacology Laboratory, Department of Pharmacy, Khwaja Yunus Ali Unviersity, as well as to Jashore University of Science and Technology and Khulna University for their valuable support. Additionally, the authors would like to thank Drug International, Bangladesh, for providing necessary chemicals, reagents, and logistic support.

## Author contributions

**Conceptualization:** Nripendra Nath Biswas, Imran Mahmud, Apurba Kumar Barman.

**Data curation:** Imran Mahmud, Md. Khalid Saifullah, Md. Niaj Morshed, Md. Arju Hossain.

**Formal analysis:** Md. Niaj Morshed, Md. Arju Hossain, Naznin Shahria, Famim Ahmed, Md. Jakaria Islam, Toufiq Ejaj Khan.

**Investigation:** Md. Khalid Saifullah.

**Methodology:** Imran Mahmud, Md. Khalid Saifullah, Md. Niaj Morshed, Naznin Shahria, Famim Ahmed.

**Project administration:** Nripendra Nath Biswas.

**Resources:** Imran Mahmud, Naznin Shahria, Toufiq Ejaj Khan.

**Supervision:** Nripendra Nath Biswas, Apurba Kumar Barman.

**Writing – original draft:** Imran Mahmud, Md. Arju Hossain, Apurba Kumar Barman, Famim Ahmed, Toufiq Ejaj Khan.

**Writing – review & editing:** Nripendra Nath Biswas, Apurba Kumar Barman, Md. Jakaria Islam.

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
