## [Decision Letter · Decision Letter 0]

27 Jun 2025

Dear Dr. Biswas,

Thank you for submitting your manuscript to PLOS ONE. After careful consideration, we feel that it has merit but does not fully meet PLOS ONE’s publication criteria as it currently stands. Therefore, we invite you to submit a revised version of the manuscript that addresses the points raised during the review process.

We look forward to receiving your revised manuscript.

Kind regards,

Rajesh Kumar Singh, Ph.D.

Academic Editor

PLOS ONE

Additional Editor Comments (if provided):

Reviewers' comments:

Reviewer's Responses to Questions

**Comments to the Author**

1. Is the manuscript technically sound, and do the data support the conclusions?

Reviewer #1: Partly

Reviewer #2: Yes

2. Has the statistical analysis been performed appropriately and rigorously?

Reviewer #1: N/A

Reviewer #2: Yes

3. Have the authors made all data underlying the findings in their manuscript fully available?

Reviewer #1: Yes

Reviewer #2: Yes

4. Is the manuscript presented in an intelligible fashion and written in standard English?

Reviewer #1: Yes

Reviewer #2: Yes

Reviewer #1: Comments

The manuscript presents data and findings that hold sufficient scientific merit and potential for publication. However, a major revision is required before it can be considered for acceptance. Notably, the current manuscript disproportionately emphasizes the in-silico analyses, whereas the title suggests a primary focus on the GC-MS profiling of C. orchioides. To maintain coherence and alignment with the manuscript’s title, greater emphasis and detailed discussion should be placed on the GC-MS findings. Additionally, the overall quality of English throughout the manuscript is suboptimal and requires substantial improvement to meet publication standards. Both the abstract and the introduction need to be comprehensively revised to enhance clarity, scientific tone, and readability. The organization and presentation of figures also requires attention. At present, there are too many figures, many of which appear redundant or unnecessarily separated. Consolidating related data into sub-panels and eliminating non-essential figures will improve the manuscript’s visual flow and overall impact.

1. The current abstract presents a concise summary of results but lacks the necessary depth and cohesion to effectively convey the study's significance. It is recommended that the authors thoroughly revise the abstract to ensure it is not only informative but also engaging, providing a clear and compelling narrative that aligns with the objectives of the research.

2. Given that traditional medicinal practices often employ aqueous extracts for plant-based remedies, the exclusive use of ethanolic extracts in this study warrants further justification. The authors should elaborate on the scientific rationale behind selecting ethanol as the primary solvent, addressing its advantages over aqueous extraction in terms of phytochemical yield, bioactive compound solubility, or other relevant factors.

3. The introduction, while comprehensive, could benefit from greater conciseness and sharper focus. Specifically, excessive details on diabetes and the enumeration of chemical compounds in Curculigo orchioides may detract from the core narrative. It is advisable to refine this section, retaining only the most pertinent information that contextualizes the study and underscores the plant's therapeutic relevance.

4. In the methodology detailing the extraction process, please specify the exact volume of 95% ethanol utilized for the maceration of 450 grams of powdered plant material. Elaborate comprehensively on the implications of this comment, addressing its relevance to extraction efficiency, solvent selection, or potential variations in phytochemical yield. A detailed response will enhance the reproducibility and scientific rigor of the study.

5. The current analysis merely reports the qualitative presence or absence of phytochemicals. To strengthen the study’s analytical depth, it is recommended to include the quantitative yield (e.g., mg/g or percentage) of each identified phytochemical per gram of ERCO. Such data would provide valuable insights into the relative abundance of different phytochemical classes, facilitating a more nuanced interpretation of their pharmacological or biological significance.

6. To further corroborate the phytochemical findings, the authors may consider employing HPLC analysis for quantitative confirmation. Utilizing optimal detection wavelengths for each class of compound would enhance the specificity and reliability of the results, aligning the study with established analytical standards in phytochemical research.

7. The figure 3 contains special characters that seems to be representing statistical analysis yet no reference has been made in the text or in figure caption to what (*, **, ****, ####, ns) these mean. Please denote correctly.

8. While the in-vitro amylase and glucosidase assays provide preliminary insights, they alone are insufficient to comprehensively evaluate ERCO's anti-diabetic potential. Incorporating cell-based assays would strengthen the mechanistic understanding of its efficacy. Furthermore, correlating these findings with in-vivo studies would yield a more robust and translational assessment.

9. The ADME-related tables should be consolidated into a supplementary file for improved readability. Instead of Table 3, a bioavailability radar plot summarizing the key pharmacokinetic properties of the selected compounds would offer a more visually intuitive representation.

10. While the molecular dynamics simulations are technically sound, the section would benefit from conciseness. A more succinct and focused presentation of the data, eliminating redundant descriptions, would enhance clarity and impact.

11. Comments with respect to the organization of figures

a. Figure 1 appears to be non-essential for the main manuscript and may be better suited for inclusion in the supplementary materials, should the editors deem it necessary.

b. Figure 2 lacks panel designations (A and B), yet the text refers to "Figure 2A" and "Figure 2B." For clarity, panel labels should be incorporated if distinct sections of the figure are being referenced. (In a figure, when we denote A and B to a certain figure, it generally refers to the panels represented in figures and not to the content of the individual figure).

c. Figures 3 and 4 would benefit from consolidation into a single figure. Additionally, the presentation of amylase and glucosidase activities should precede the in vivo experimental results to maintain a logical flow.

d. Figures 5, 6, 7 and 8 should be reorganized into a square or grid-like format rather than a linear arrangement to enhance visual coherence and optimize space utilization.

e. Figures 9, 10, and 11 could be compiled into a single composite figure for a more streamlined presentation. A recommended reference for organizing molecular dynamics simulation results is the article "In-silico evidence of ADAM metalloproteinase pathology in cancer signaling networks," which provides an effective model for such consolidation.

Reviewer #2: The researchers have conducted an acute toxicity study, and the results have been included in the manuscript. However, since the crude extracts were administered to the mice, it would be beneficial to also incorporate chronic toxicity study findings. Additionally, including kidney and liver function test results would further strengthen the manuscript.

**Do you want your identity to be public for this peer review?** For information about this choice, including consent withdrawal, please see our Privacy Policy

Reviewer #1: No

Reviewer #2: No

---

## [Author Response · Author response to Decision Letter 1]

10 Aug 2025

Pls see the attached "response to reviewer's" file

---

## [Decision Letter · Decision Letter 1]

13 Oct 2025

GCMS profiling of bioactive phytocompounds from Curculigo orchiodesGaertn. Root extract and evaluation of antioxidant and antidiabetic activities: A computational drug development approaches

PONE-D-25-24094R1

Dear Dr. Biswas,

We’re pleased to inform you that your manuscript has been judged scientifically suitable for publication and will be formally accepted for publication once it meets all outstanding technical requirements.

Kind regards,

Rajesh Kumar Singh, Ph.D.

Academic Editor

PLOS ONE

Additional Editor Comments (optional): Improved

Reviewers' comments:

Reviewer's Responses to Questions

**Comments to the Author**

Reviewer #3: All comments have been addressed

Reviewer #4: All comments have been addressed

2. Is the manuscript technically sound, and do the data support the conclusions?

Reviewer #3: Yes

Reviewer #4: Yes

3. Has the statistical analysis been performed appropriately and rigorously?

Reviewer #3: Yes

Reviewer #4: I Don't Know

4. Have the authors made all data underlying the findings in their manuscript fully available?

Reviewer #3: Yes

Reviewer #4: Yes

5. Is the manuscript presented in an intelligible fashion and written in standard English?

Reviewer #3: Yes

Reviewer #4: Yes

Reviewer #3: The Authors responded to the previous comments point by point, and most of the substantive comments were incorporated. In several places, justification was provided for retaining the original solution, which is acceptable. I believe the manuscript in its current form is suitable for publication.

Reviewer #4: The authors have thoroughly and comprehensively addressed all the comments and concerns raised by the reviewers. Each point highlighted in the review has been carefully considered, and appropriate modifications or clarifications have been made throughout the manuscript to ensure that the revised version meets the standards required for publication.

**Do you want your identity to be public for this peer review?** For information about this choice, including consent withdrawal, please see our Privacy Policy

Reviewer #3: **Yes: ** Radosław Kowalski

Reviewer #4: No

---

## [Editor Report · Acceptance letter]

PONE-D-25-24094R1

PLOS ONE

Dear Dr. Biswas,

I'm pleased to inform you that your manuscript has been deemed suitable for publication in PLOS ONE. Congratulations! Your manuscript is now being handed over to our production team.

Kind regards,

on behalf of

Dr. Rajesh Kumar Singh

Academic Editor

PLOS ONE